# Miconazole Nitrate Microparticles in Lidocaine Loaded Films as a Treatment for Oropharyngeal Candidiasis

**DOI:** 10.3390/ma16093586

**Published:** 2023-05-07

**Authors:** Guillermo Tejada, Natalia L. Calvo, Mauro Morri, Maximiliano Sortino, Celina Lamas, Vera A. Álvarez, Darío Leonardi

**Affiliations:** 1Grupo Materiales Compuestos Termoplásticos, Instituto de Investigaciones en Ciencia y Tecnología de Materiales, Av. Colón 10850, Mar Del Plata 7600, Argentina; 2Instituto de Química Rosario, Suipacha 570, Rosario 2000, Argentina; 3Área de Análisis de Medicamentos, Departamento Química Orgánica, Facultad de Ciencias Bioquímicas y Farmacéuticas, Universidad Nacional de Rosario, Suipacha 570, Rosario 2000, Argentina; 4Planta Piloto de Producción de Medicamentos, Facultad de Ciencias Bioquímicas y Farmacéuticas, Universidad Nacional de Rosario, Suipacha 570, Rosario 2000, Argentina; 5Centro de Referencia de Micología, Área Farmacognosia, Departamento Química Orgánica, Facultad de Ciencias Bioquímicas y Farmacéuticas, Universidad Nacional de Rosario, Suipacha 570, Rosario 2000, Argentina; 6Área Farmacognosia, Departamento Química Orgánica, Facultad de Ciencias Bioquímicas y Farmacéuticas, Universidad Nacional de Rosario, Suipacha 570, Rosario 2000, Argentina; 7Área Técnica Farmacéutica, Departamento Farmacia, Facultad de Ciencias Bioquímicas y Farmacéuticas, Universidad Nacional de Rosario, Suipacha 570, Rosario 2000, Argentina

**Keywords:** oropharyngeal candidiasis, miconazole nitrate, lidocaine, polymeric microparticles, polymeric films

## Abstract

Oral candidiasis is an opportunistic infection that affects mainly individuals with weakened immune system. Devices used in the oral area to treat this condition include buccal films, which present advantages over both oral tablets and gels. Since candidiasis causes pain, burning, and itching, the purpose of this work was to develop buccal films loaded with both lidocaine (anesthetic) and miconazole nitrate (MN, antifungal) to treat this pathology topically. MN was loaded in microparticles based on different natural polymers, and then, these microparticles were loaded in hydroxypropyl methylcellulose-gelatin-based films containing lidocaine. All developed films showed adequate adhesiveness and thickness. DSC and XRD tests suggested that the drugs were in an amorphous state in the therapeutic systems. Microparticles based on chitosan-alginate showed the highest MN encapsulation. Among the films, those containing the mentioned microparticles presented the highest tensile strength and the lowest elongation at break, possibly due to the strong interactions between both polymers. These films allowed a fast release of lidocaine and a controlled release of MN. Due to the latter, these systems showed antifungal activity for 24 h. Therefore, the treatment of oropharyngeal candidiasis with these films could reduce the number of daily applications with respect to conventional treatments.

## 1. Introduction

Candidiasis is one of the most common human opportunistic fungal infections of the oral cavity. Candida strains reside as a commensal in oral flora; however, it causes fungal infection when the host becomes immunocompromised. Particularly, oral candidiasis is common in the population of patients with hematological malignancies as well as in people with acquired immunodeficiency syndrome (AIDS) [1]. Currently, there are several antifungal drugs (in different presentations) approved for the topical treatment of oral candidiasis [2], among them are ketoconazole gel 2% (dose: 3 times/day), clotrimazole gel 1% (dose: 3 times/day), nystatin suspension (dose: 4–6 mL/6 h) and miconazole nitrate (MN) gel 2% (dose 100 mg/6 h). These formulations allow the eradication of fungal infection but need to be applied several times daily, and the treatment in HIV/AIDS patients may be prolonged (from 4 to 14 days using nystatin rinses) [3]. Thus, the development of an appropriate formulation to treat oropharyngeal candidiasis, which allows for a reduction in the number of applications per day, will improve patient’s compliance during treatment. In recent years, several buccal devices containing MN have been developed and analyzed. In 2014, Rai et al. developed a cellulosic polymer-based gel containing MN for buccal delivery. The final formulation, obtained after an optimization procedure, showed an extended residence time in oral mucosa and a broader zone of growth inhibition compared with a marketed antifungal formulation [4]. However, in general, gels have a short residence time on the mucosa, which is a disadvantage that can be solved when buccal films are developed [5]. In 2018 Mady et al. developed polymeric films containing MN for treatment of buccal candidiasis. The authors combined MN with urea in the films and observed an improvement in the inhibition zone diameters for films containing increasing concentrations of both urea and MN. It was concluded that urea acts as a penetration enhancer against *C. albicans* and there is a synergistic effect between MN and urea. Although this finding is highly relevant, the release of the MN from the films was analyzed only for 120 min, and therefore, it is not possible to determine the number of applications per day that will be necessary for treatment [6]. On the other hand, in 2021, De Caro et al. developed and characterized solid and semisolid formulations containing MN-loaded solid lipid microparticles as therapy for oral candidiasis. The developed formulations allowed them to overcome the short retention time and suboptimal drug concentration that are present in the currently available antifungal therapy. The solid lipid microparticles loaded with MN were able to increase up to three-fold MN accumulation in the buccal mucosa compared with Daktarin^®^ 2% oral gel, probably due to the effect of oromucosal penetration enhancers of the microparticles. Finally, these solid lipid microparticles were loaded into a buccal gel (based on trehalose, PVP-K90, and hydroxyethylcellulose) or into a mucoadhesive buccal film based on trehalose, PVP-K90, limonene, and hydroxyethylcellulose. In particular, the buccal film was able to release the MN for an extended period of time and avoid MN permeation, limiting the possibility of adverse side effects [7]. These findings are remarkable and allow for progress in the search for an ideal device for treating buccal candidiasis. Since candidiasis may cause burning pain and pruritus, it is desirable to develop a formulation containing an anesthetic in addition to an antifungal agent. Thus, the purpose of this work was to develop films loaded with both lidocaine (to obtain a fast dissolution rate of the anesthetic) and miconazole loaded microparticles (to achieve controlled release of the antifungal agent) and evaluate the antifungal activity over time in the systems.

## 2. Materials and Methods

### 2.1. Materials

Chitosan (CH, MW 230 KDa and 80.6% N-deacetylation) was acquired from Aldrich Chemical Co. (Milwaukee, WI, USA) and hydroxypropyl methylcellulose (HPMC, MW 250 kDa, hydroxypropyl 7–12%, methoxyl 19–24%) was obtained from Eigenmann & Veronelli (Milan, Italy). Gelatine (GEL, type A from pork skin, 125 Bloom value) and sodium alginate (ALG, Sigma-Aldrich Co. Buenos Aires, Argentina) were also used. FMC BioPolymer (Philadelphia, PA, USA) donated samples of food-grade λ and κ-carrageenan (λ-c and κ-C), and sodium lauryl sulphate (SLS) was acquired from Biopack (Buenos Aires, Argentina). Ammonium acetate (HPLC grade) was acquired from Thermo Fisher Scientific (Cleveland, OH, USA), acetonitrile (HPLC grade) and sorbitol (70%) were obtained from Cicarelli (Buenos Aires, Argentina), while lidocaine hydrochloride (LDCH) and miconazole nitrate (MN), both of pharmaceutical grade, were obtained from Parafarm (Buenos Aires, Argentina). All other reagents were of analytical grade.

### 2.2. Preparation of Microparticles

CH solutions (0.5% *w*/*v*) were obtained by dispersing the polymer in acetic acid solution (pH = 2.50, 30% *v*/*v*), while SLS, ALG, κ, and λ-carrageenan solutions (0.2% *w*/*v*) were prepared by dissolving them in water. The solutions were stirred at 400 rpm and 40 °C for 30 min). Subsequently, MN at 20% w/w (with respect to the mass of polymers) was added to the solution and stirred again at 400 rpm for 10 min at 40 °C. Then, mixtures were sprayed using a Buchi Mini dryer B-290 (Flawil, Switzerland). The composition of the microparticles was 50% CH and 50% SLS, ALG, κ or λ-carrageenan. Some parameters such as airflow rate (38 m^3^/h), feed rate (5 mL/min), pump (10%), aspirator (100%), spray-drying (SD) inlet temperature (130 °C), and outlet temperature (70 °C) remained constant during the prosses. After SD, the powders were stored at room temperature.

### 2.3. Lidocaine Films Loaded Microparticles

Solutions of HPMC and GEL both at 3% *w*/*v*, were obtained by dispersing the polymers in water. The dispersion was stirred (for 24 h) and filtered using Miracloth^®^ (Calbiochem-Novabiochem Corp., San Diego, CA, USA). Then, GEL solution was dripped over the HPMC one under stirring (400 rpm at 80 °C). Sorbitol at 20% w/w (as a plasticizer) and LDCH (at 5% *w*/*w* with respect to the mass of polymers) were added to the mixture. This mixture was stirred again at 400 rpm for 30 min. Then, different microparticulate systems loaded with MN were added to the mixture (at 2% *w*/*w* of the total polymeric mass) and stirred at 400 rpm for 1 h at 50 °C to homogenize). Dispersions were poured into Petri dishes (9 cm diameter) and dried at 40 °C and 58% relative humidity (RH) for 48 h. After that, films were detached from the Petri dishes and conditioned for 72 h (25 °C and 58% RH). Systems presenting lack of physical defects such as cracks, holes, and bubbles were selected and used in the different tests. The different compositions of the systems are shown in Table 1.

### 2.4. High-Performance Liquid Chromatography (HPLC)

High-Performance Liquid Chromatography was carried out using an Agilent Technologies 1200 Series chromatograph (Santa Clara, CA, USA) comprising a SiliaChrom C18 column (5 μm, 150 ×  4.6 mm, S/N S11410) thermostatted at 40 °C [8] and a diode array detector. The amounts of MN and LDCH released were obtained spectrophotometrically by measuring the absorbance at 230 [9] and 265 nm [10] using ammonium acetate buffer (50 mM, pH 4): acetonitrile (35:65) as the mobile phase in isocratic mode with a flow rate of 1.5 mL min^−1^. The retention time of LDCH was 2.9 min while that of MN was 14.4 min. The construction of the LDCH calibration curve was developed in the range 25–400 mg/mL (equation: y = 1675x + 80,010) showing R^2^ = 0.9795, while for MN the range was 20–400 mg/mL (equation: y = 41,953x + 185,070) showing R^2^ = 0.998.

### 2.5. Microparticles Loaded MN—Encapsulation Efficiency

Encapsulation efficiency (EE) of each microparticle system was obtained using the HPLC method described above. For this, 12 mg of microparticles was dispersed in 60 mL of methanol and the suspension was stirred for 2 h at 400 rpm to extract the MN from the microparticles.

### 2.6. Films Loaded Microparticles: Thickness and Folding Endurance

A digital micrometer (Schwyz, China) was used to determine the thickness (TH) of the films. Six measurements were carried out in the center and around of each system. Folding endurance values were obtained by repeatedly folding each system 300 times at the same place, or until it broke. The assay was carried out in triplicate [11].

### 2.7. Mechanical Properties

The mechanical strength of the systems was analyzed by using a Universal Testing Machine Instron, Series 3340, single column (Instron, Norwood, MA, USA) with a 50 N load cell. Systems were conditioned (24 h at 25 °C and 58% RH) for each mechanical test, and then cut into strips 7 mm wide and 60 mm long, to evaluate both tensile strength and elongation at break. To carry out the assay it was settled at 30 mm the initial grip distance and at 0.05 mm/s the crosshead speed. From stress/strain curves, the parameters tensile strength and elongation at break were obtained. The assay was carried out in triplicate [12].

### 2.8. In Vitro Mucoadhesive Strength

The in vitro mucoadhesive strength was obtained following the technique described by Tejada et al. [13]. Briefly, an Instron universal testing machine was employed to analyze the mucoadhesive strength of each system. This parameter was evaluated in vitro by determining the force required to detach each system from a pork gum disc (donated by “Paladini” slaughterhouse, V.G. Galvez, Argentina)

### 2.9. Swelling Index

Swelling index (SI) values were obtained by immersing a portion of the systems (2.5 cm diameter) in artificial saliva (1 mL, 37 °C, pH = 6.8). At different time intervals, the systems were removed, and the excess adhering moisture was gently blotted off and weighed. Then, artificial saliva was added again (0.5 mL), and the process was repeated until 24 h. Using the Eqaution (1), the SI was calculated. The assay was carried out in triplicate.
(1)$$SI=\left(W_{t}-W_{0}\right)/W_{0}$$
where *W_t_* is the weight of swollen films and *W*_0_ is the weight of dry film.

### 2.10. Morphology Analysis and Size Determination by Scanning Electron Microscopy

Scanning electron microscopy (SEM, AMR 1000, Leitz, Wetzlar, Germany) was used to observe the morphology of the systems. Both microparticles and films containing microparticles were placed on an aluminum sample holder using a conductive double-sided adhesive. To make the systems conductive they were coated with a fine gold layer (15 min at 70–80 mTorr). SEM images were obtained using 20 kV accelerating voltage and magnifications of 200× and 500× for films and 10,000× for microparticles.

### 2.11. Thermal Analysis

Thermogravimetric analysis (TGA) was performed using a TG HI-Res thermal analyzer (TA Instruments) at a 10 °C/min heating rate in the range of room temperature to 800 °C, in air flow. Differential scanning calorimetry (DSC) tests were carried out in a DSC Q2000 (TA Instruments) at a 10 °C/min heating rate in the range of room temperature to 200 °C, under nitrogen.

### 2.12. X-ray Diffraction

An automated X’Pert Phillips MPD diffractometer (Eindhoven, The Netherlands) was used, and data collection was performed in transmission mode. The patterns of X-ray diffraction (XRD) were recorded using CuKα radiation (λ = 1.540562 Å), 20 mA (current), 40 kV (voltage), and 0.02° (steps) on the interval 2θ from 10 ° to 50 °. The Stoe Visual-Xpow package, Version 2.75 (Germany) was employed for data acquisition and evaluation.

### 2.13. Dissolution Studies

Dissolution assays were performed in artificial saliva (900 mL, 37 °C), using a USP XXIV apparatus II (Hanson Research, SR8 8-Flask Bath, ON, Canada) with paddles rotating at 50 rpm [10,14]. Portions of films (containing 25 mg LDCH) were placed in the dissolution medium. At different times, 3 samples of 1 mL each were taken, and after each sample collection, an equal volume of the dissolution medium was added. The drug content of the aliquots was determined using HPLC as described above [15].

### 2.14. Halo Zone Test over Time

The guidelines of the disk diffusion method described in CLSI document M44-A2 were followed to carry out the halo zone test [16]. Agar plates (90 mm diameter) containing Mueller—Hinton agar, supplemented with glucose and methylene blue, were used to perform the assay. The agar surface was inoculated by dipping sterile cotton swabs into a fungal suspension and by streaking the plate surface in three directions. After the plate was dried for 20 min, systems (containing 10 mg of MN) were placed onto the surface of the agar. Each hour, the films were moved to another zone of the culture plate and the process was repeated for 24 h. A paper disk containing MN powder (10 mg) was designed as a control. Finally, the plates were incubated in air (28 °C) and read after 24 h. Using a caliper, the zone of complete growth inhibition (halo diameters) was determined [17]. This assay makes it possible to analyze the release of MN from the systems and their antifungal activity over time.

### 2.15. Statistical Analysis

An ANOVA test was used, and when it was found that the effect of the factors was significant, the Tukey multiple ranks test was applied to establish significant differences among samples with a 95% confidence (*p* < 0.05).

## 3. Results and Discussion

### 3.1. Encapsulation Efficiency of Microparticles

Encapsulation efficiencies values ranged from 48.88% to 97.15%. The highest encapsulation value corresponds to the CH-ALG system, while the lowest value was obtained in the microparticles based on CH-SLS. Systems formed by CH-κC and CH-λC presented intermediate encapsulation values: 67.37 and 57.44%, respectively. This could be because, before the SD process, polymers with a high charge density have a greater number of free sites to interact with the drug, increasing its capacity to encapsulate the MN [18]. Furthermore, the EE values are also probably related to the positive charge densities present in CH and the negative charge densities present in the other polymers. In the preparation of the CH solution (pH = 2.50), the amino groups are positively charged, while the other polymers (pH solution = 5.40) are negatively charged. ALG has in its structure monomeric units of α-L-guluronic acid, β-D-mannuronic units, and the two bonded together, being a polymer with abundant negative charge density which may interact more easily with CH [19]. Following the descending order of EE values, the Carrageenans present sulfate ester groups in the galactose units to interact with the amino groups of the CH [20]; finally, the lowest EE value corresponds to the CH-SLS formulation, which could be due to the fact that in the SLS chain there is only one sulfate group to interact with CH.

### 3.2. Film Thickness, Folding Endurance and Mechanical Properties

Once the films were loaded with microparticles, film TH, folding endurance, and mechanical properties were determined. TH measurements are essential to evaluate the potential discomfort of the patient in the gingiva. The TH values of the formulations ranged from 0.527 to 0.820 mm with no significant differences (*p* > 0.05) between them (Table 2). These TH values are lesser than 1 mm and therefore are considered adequate to avoid discomfort after application [21].

The folding endurance is the number of times each film can be folded in the same place without breaking. The analysis was carried out to analyze the flexibility of the films so that they can be easily manipulated, without causing discomfort, and achieve a safe application at the site of action [11]. All formulations were folded at least 300 times, complying with the test (Table 2). The obtained values of elongation at break and tensile strength for the different films are also shown in Table 2. High tensile strength values were observed in the films containing CH-ALG and CH-κC microparticles. Although the matrix of all four polymeric films is constituted by the same polymers, the difference lies in the loaded microparticulate system. Those microparticles that have higher charge density and free charges would interact strongly with the polymers in the films.

The TH of the films varied with respect to the values obtained for the not loaded film (0.517 mm ± 0.04), demonstrating that the incorporation of the microparticles produced a slight increase in the TH of the formulations. This result is probably because the presence of microparticles disrupts the polymeric matrix of the films, preventing an adequate compaction, which is also reflected in the tensile strength and elongation values of the formulations [22,23,24]. The tensile strength of the not loaded film (3.28 N) was diminished significantly with the incorporation of microparticles. When the microparticles based on CH-ALG and CH-κC were added, the tensile strength of the films was reduced from 3.28 N to 2.30 N and 2.20 N, respectively. Additionally, the incorporation of the microparticles based on CH-λC and CH-SLS reduced even more this parameter, reaching values of 1.53 N and 1.36 N, respectively. In addition, the elongation of films containing microparticles based on CH-ALG, CH-κC, and CH-λC decreased significantly with respect to the not load film (18.54 ± 4.73%). Conversely, this parameter was not altered when the film was loaded with the CH-SLS microparticles. As mentioned, CH is a cationic polymer presenting amino groups, which can interact with anionic and neutral compounds, while ALG, C, and SLS are anionic polymers which present different groups (carboxylic, sulphate ester, and sulphate groups, respectively) to interact with the amino groups of CH. Yuhua C. et al. reported that the reactivity of the substituent anionic groups of the mentioned anionic polymers increases in the following order: (1) sulfate group, (2) sulfate ester group, (3) carboxylic group [25]. Thus, the microparticles based on CH and ALG present the strongest interactions [26]. Between the loaded films, those containing CH-ALG microparticles were more rigid while those loaded with CH-SLS showed the smallest tensile strength and the highest elongation at break. This fact shows a direct relationship between the interactions that exist in the microparticles with the mechanical properties of the loaded films.

### 3.3. Swelling Index

The swelling index values are shown in Figure 1. The formulations loaded with CH-ALG, CH-κC, and CH-λC microparticles were able to complete the test after 24 h. Systems loaded with CH-SLS and CH-λC microparticles achieved maximum in swelling at approximately at 6 h assay. After this time, the weight of both films decreased due to the partial (CH-λC) and complete (CH-SLS) disintegration of the matrices. Polymers presenting free groups with hydrophilic nature play an important role in water uptake [27]. In film containing microparticles that present strong interactions between their oppositely charged polymers (CH-ALG), the amino groups of CH interact with the carboxylic groups of ALG. These interactions reduced the number of amino and carboxylic free groups, diminishing the water retention capacity of this matrix. Oppositely, microparticles based on CH-SLS are able of capturing a greater amount of fluid, generating greater swelling, which finally causes the disintegration of the matrix.

### 3.4. In Vitro Mucoadhesive Strength

Mucoadhesivity values of the systems are shown in Figure 2. Films loaded with microparticles significantly decreased the adhesiveness of the unloaded film (1.52 N ± 0.18). On the other hand, no significant differences were found between the adhesiveness of the different loaded films. This fact could be due to the presence of the microparticles; when they interact with the matrix, the number of free surface charges available to interact with the mucin of the gingiva is reduced [28]. The films loaded with microparticles based on SLS were the most flexible, presenting the highest swelling index, so they could adhere more easily to the gingiva than the other formulations [29].

### 3.5. Morphology Analysis and Size Determination by Scanning Electron Microscopy

Images taken by SEM of the microparticulate systems and the films loaded with microparticles are shown in Figure 3. CH-ALG microparticles (Figure 3a) showed a smooth and uniform surface (size 2.973 ± 0.886 μm), while microparticles based on CH-κC and CH-λC (Figure 3b,c) presented nonuniform and irregular surface (size 2.239 ± 0.574 and 1.734 ± 0.382 μm, respectively), whereas microparticles based only on CH-SLS (size 2.774 ± 1.035 μm) were irregular with a smooth surface (Figure 3d). All films loaded with microparticles showed a nonhomogeneous surface with cavities, while the presence of discontinuous areas and rugosities were observed in the cross section of films (Figure 3i–l). A probable incompatibility between HPMC and GEL has previously been reported, which may be the cause of this morphology [30].

### 3.6. Thermal Characterization

Figure 4 shows the derivate of TGA curves of the systems. Regarding the MN (Figure 4a), it has a broad peak centered at 305 °C corresponding to its degradation [13]. In the microparticles loaded with MN, the characteristic peaks of the degradations step of the forming polymers were observed, but not the peaks of MN [31,32,33,34,35]. As observed in Figure 4b, LDCH presented a degradation peak centered at around 250 °C [36]. The different film formulations exhibit a central peak around 300 °C, corresponding to the degradation of the film forming polymers, HPMC, and GEL [37]. The absence of the characteristic peaks of MN and LDCH loaded in the systems is due to the ability of the polymeric matrices to protect both drugs against thermal degradation [38].

The melting temperatures of the systems were observed by DSC (Figure 5). As shown in Figure 5a, the endothermic peak for MN was at 186 °C [39]. This endothermic peak was absent in MN loaded microparticles. Figure 5b shows the endothermic peak of LDCH at 80 °C, corresponding to the fusion of this drug [40]. As in the case of MN, the same performance of the drug in the different formulations was observed in this curve. The absence of the characteristic melting peaks of MN and LDCH is due to: (1) the drug reduced its crystallinity below the detection limit, or (2) the drug changed to an amorphous state [41,42].

### 3.7. X-ray Diffraction

Figure 6a shows the diffractogram of MN and microparticles loaded with MN and Figure 6b shows the patterns of LDCH and films loaded with the microparticles and LDCH. The diffractogram of the MN presented the typical crystalline pattern [43]. On the other hand, the XRD patterns of the loaded microparticles did not show any defined peak. This could be due to multiple factors such as the fact that, in the SD process, the MN partially lost its crystallinity or, that the MN loaded in the microparticulate systems is in an amorphous state [42].

As observed in Figure 6b, the diffractogram of the LDCH showed peaks at diffraction angles (2θ): 6.9 °, 14.2 °, 20.8 °, 25.2 °, and 27.8 °, with a typical crystalline pattern [44]. The characteristic main peaks of LDCH were absent in the XRD pattern of the films, suggesting that LDCH was in an amorphous state.

### 3.8. Dissolution Studies

The dissolution of LDCH (as raw material), and the different systems in artificial saliva are shown in Figure 7. LDCH raw material showed a fast dissolution rate (100% after 1 h assay), while when the drug was loaded in the films, its dissolution rate was reduced. The release of LDCH from the films was in agreement with the swelling results. Among the formulations, films containing microparticles based on CH-SLS allowed for the fastest dissolution rate of the drug (60% after 1 h assay), while films containing microparticles based on CH-ALG released only 20% of LDCH at this time. This result was predictable, since LDCH is a drug that is highly soluble in water; the more swelling the matrix presents, the greater the dissolution of the drug there will be.

### 3.9. Halo Zone Test over Time

All systems produced different inhibition halos at different times (Figure 8).

After 1 h assay, the highest inhibition halo corresponds to MN powder (4.2 cm) followed by the formulations HG-CH-SLS, HG-CH-κC, and HG-CH-λC (around 3.6 cm). This could be due to the fact that the interactions in the microparticles loaded in these films are weaker than those generated in the CH-ALG microparticles. Thus, the MN is more exposed to the culture medium, resulting in higher inhibition halos at the beginning of the study. Contrarily, after 1 h assay, the formulation HG-CH-ALG produced the smallest halo (2.6 cm). In the same way, both the strongest interaction between CH and ALG and the reduced swelling of this film probably cause less exposition of MN to the culture medium. The activity of the systems was decreasing over time, the paper disk containing MN powder lost its activity after 7 h assay, followed by the film HG-CH-SLS which did not show any halo after 8 h assay, and then HG-CH-κC (which lost its activity after 12 h). On the other hand, the formulation HG-CH-λC presented activity for 12 h, and finally, the film containing microparticles based on CH-ALG showed continuous antifungal activity until 24 h assay, demonstrating that this system (containing MN microencapsulated and added to the HPMC-GEL matrix) produced a sustained release of this drug.

## 4. Conclusions

In this work, microparticulate systems containing miconazole nitrate were developed by the spray-drying method. These systems were loaded in films based on HMPC and GEL which also contain lidocaine as an anesthetic drug to complement the treatment of oropharyngeal candidiasis. The MN encapsulation efficiency was higher in the CH-ALG system, possibly due to the high reactivity of the amino and carboxylic groups of CH and ALG, respectively, which may interact with the drug. These strong interactions between CH-ALG generated films with the highest tensile strength and the lowest elongation at break. All systems presented adequate values in terms of thickness to not produce discomfort to the gingiva of the potential patient. In the DSC and XRD tests, the characteristic peaks of the drugs were not evident, possibly because the drugs changed to an amorphous state in the systems. In vitro activity of the formulations evaluated by the halo zone test showed that the formulation containing microparticles based on CH-SLS (which allowed the fastest dissolution of MN) lost its antifungal activity after 8 h, while the formulation containing CH-ALG microparticles presented activity for 24 h. Therefore, the film loaded with both lidocaine and microparticles based on CH-ALG containing MN would cause an anesthetic effect and produce a controlled release of MN in the patient’s gums for 24 h. Thus, this formulation could be applied once a day, increasing the patients’ compliance.

## Figures and Tables

**Figure 1 materials-16-03586-f001:**
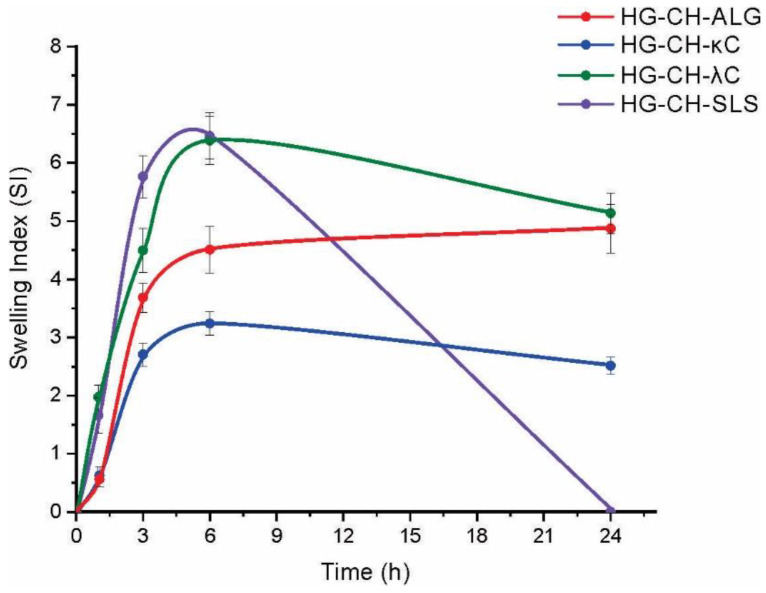
Swelling index of different systems in artificial saliva.

**Figure 2 materials-16-03586-f002:**
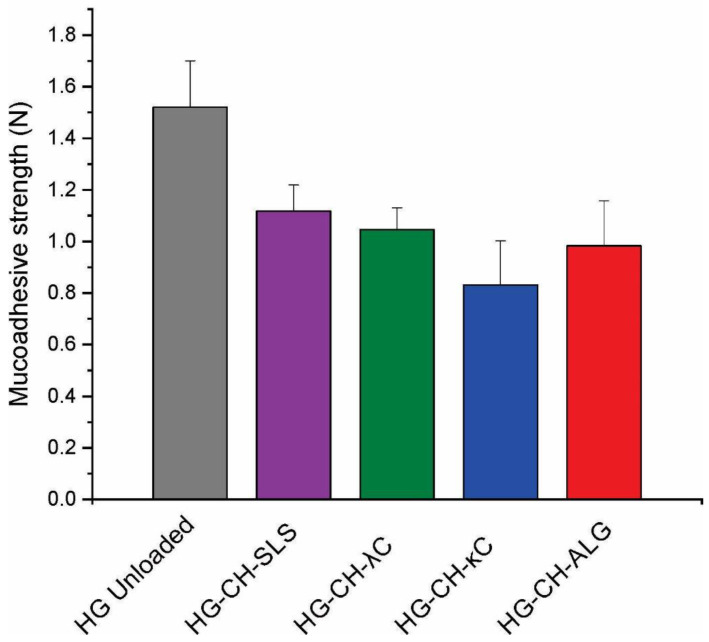
Film mucoadhesive strength values.

**Figure 3 materials-16-03586-f003:**
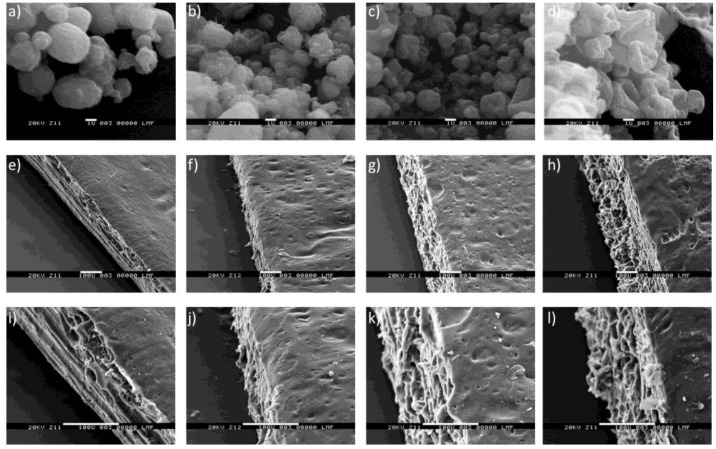
Scanning electron microscopy of microparticles (**a**–**d**) at 10,000× and films loaded with microparticles at 200× (**e**–**h**) and at 500× (**i**–**l**). (**a**) CH-ALG, (**b**) CH-κC, (**c**) CH-λC, (**d**) CH-SLS, (**e**) HG-CH-ALG, (**f**) HG-CH-κC, (**g**) HG-CH-λC, (**h**) HG-CH-SLS, (**i**) HG-CH-ALG, (**j**) HG-CH-κC, (**k**) HG-CH-λC, (**l**) HG-CH-SLS.

**Figure 4 materials-16-03586-f004:**
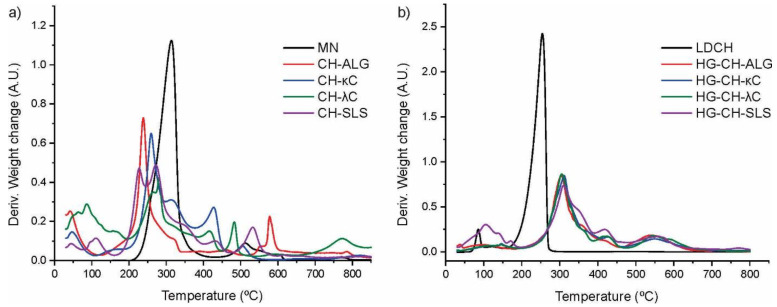
(**a**) DTGA curves of MN and microparticles loaded with MN. (**b**) DTGA curves of LDCH and different film formulations.

**Figure 5 materials-16-03586-f005:**
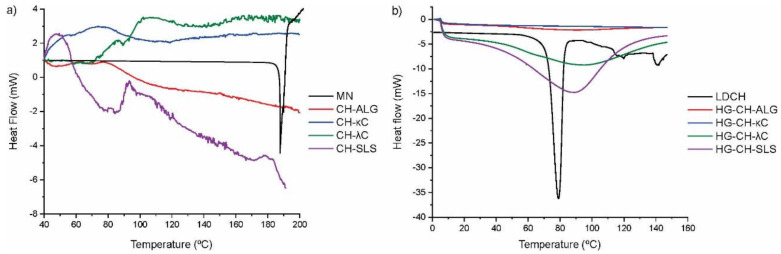
(**a**) DSC curves of MN and microparticles loaded with MN. (**b**) DSC curves of LDCH and films loaded with microparticles and LDCH.

**Figure 6 materials-16-03586-f006:**
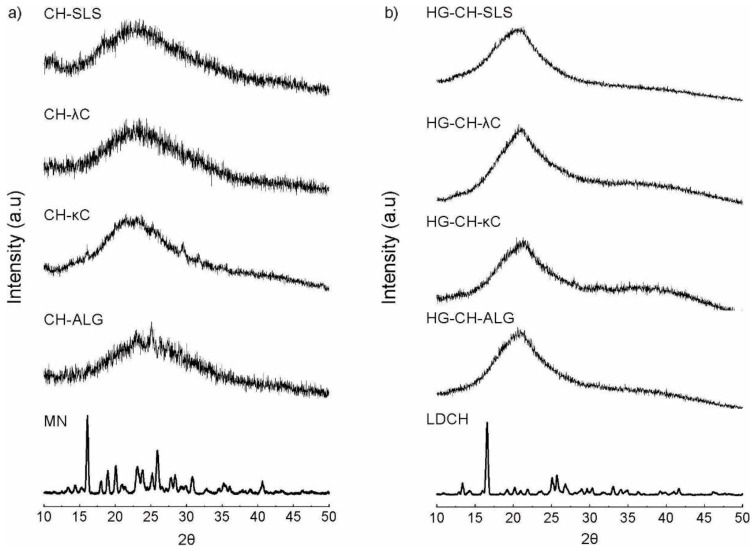
(**a**) Diffractogram of MN and microparticles loaded with MN. (**b**) Diffractogram of LDCH and films loaded with microparticles and LDCH.

**Figure 7 materials-16-03586-f007:**
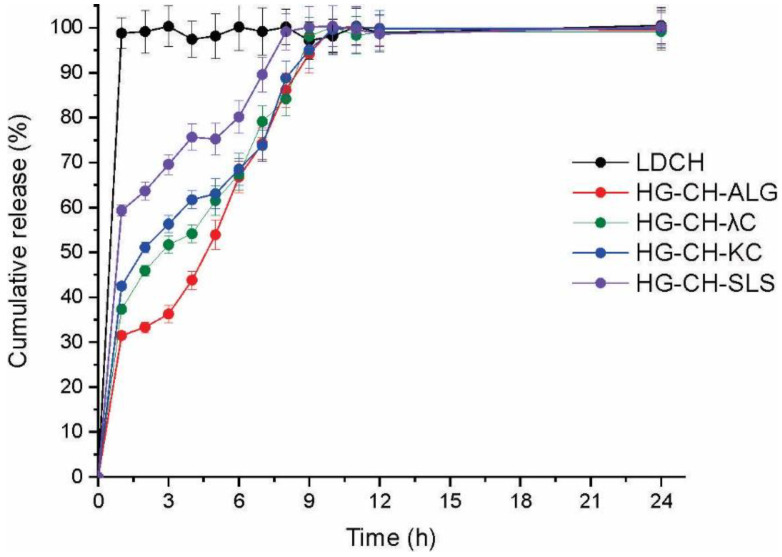
Dissolution profiles of LDCH raw material and films loaded with microparticles and LDCH.

**Figure 8 materials-16-03586-f008:**
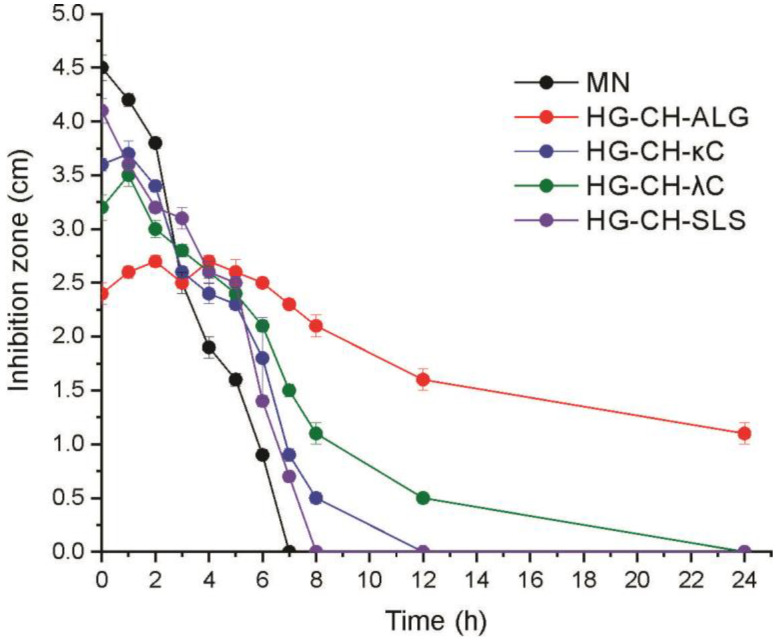
Inhibition halos produced by all formulations in cultures of *C. albicans*.

**Table 1 materials-16-03586-t001:** Composition of different formulation.

Films Matrix Composition (Containing LDCH 2% *w*/*w*)	Microparticles Matrix Composition (Containing MN 5% *w*/*w*)	Films Loaded Microparticles Abbreviation
HPMC-GEL-LDCH	CH-ALG-MN	HG-CH-ALG
HPMC-GEL-LDCH	CH-κC-MN	HG-CH-κC
HPMC-GEL-LDCH	CH-λC-MN	HG-CH-λC
HPMC-GEL-LDCH	CH-SLS-MN	HG-CH-SLS

**Table 2 materials-16-03586-t002:** Values obtained in physical measurements and mechanical properties.

Film Formulation	Thickness (mm)	Folding Endurance	Tensile Strength (N)	Elongation (%)
HG-CH-ALG	0.818 ± 0.085	**√**	2.30 ± 0.57	12.11 ± 3.37
HG-CH-κC	0.696 ± 0.118	**√**	2.20 ± 0.54	15.22 ± 5.07
HG-CH-λC	0.820 ± 0.075	**√**	1.53 ± 0.33	14.97 ± 5.60
HG-CH-SLS	0.527 ± 0.076	**√**	1.36 ± 0.48	19.04 ± 3.92

## Data Availability

The data presented in this study are available on request from the corresponding authors.

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
