# Peer review of "Miconazole Nitrate Microparticles in Lidocaine Loaded Films as a Treatment for Oropharyngeal Candidiasis"

_materials, 2023, doi:10.3390/ma16093586_

Round 1

Reviewer 1 Report

The authors expose an interesting work on the formulation and characterization of buccal films containing Miconazole and Lidocaine. The results seem promising. Some criticisms, however, emerge from the reading of the text.

The research objectives and the design of project are good but the manuscript has too many gaps and needs to be implemented with more scientific rigor.

Abstract:

The abstract must be entirely rewritten because it presents serious grammatical and content gaps. Indeed, it lacks the characterizations and/or the main results described in the manuscript. Many sentences are grammatically incorrect or truncated.

Introduction:

Line 50, it might be useful to add the microparticulate systems (i.e. 10.3390/pharmaceutics13091361)

Reference 5 is too old. There are much more recent examples in the literature. For example: 10.3390/pharmaceutics11010035; 10.1016/j.jconrel.2022.10.058

From Line 55 to line 85  the authors summarized the results of five of their previous works. Both the details of the results described and the way of summarizing them cannot be the only aspect reported in the introduction of a manuscript. As a consequence, from references 6 to 10, they are all self-citations. The introduction needs to be entirely rewritten with additional argumentation.

Materials

It would be appropriate in this paragraph to put the name of the substances in full, followed by the acronym.

The brands of the solvents used and their degree of purity are missing.

Other components as Sorbitol are missing.

Is LDCH an acronym for Lidocaine? It is never indicated throughout the manuscript.

Methods:

In paragraph: Preparation of microparticles

The percentage of CH compared to that of SLS, ALG, κ or λ-carrageenan in the composition of the microparticles matrix is missing or not clearly expressed.

Line 114: there is confusion about the acronym "GEL". Does it refer to gelatine alone or to CH-gelatin as expressed in the introduction?

In paragraph: Microparticles loaded MN - Encapsulation Efficiency 

The Detector of HPLC system has been omitted as well as if the HPLC method is in gradient or in isocratic.

Miconazole has a maximum of UV absorption at λ = 272 nm, why did the authors quantify it at λ = 230 nm?

folding endurance: is missing on how many films have been tested (reproducibility)

In paragraph: Dissolution Studies

The Detector of HPLC system has been omitted

The retention time of MN and LDCH have been omitted.

The calibration curves to quantify MN and LDCH has been omitted

Results and discussion paragraph is well argued but needs an English language revision.

Conclusion paragraph should be improved and made more clear to be read

Author Contributions are missing

Author Response

We wish to thank the reviewer; his/her suggestions improved the quality of this manuscript. A point-by-point response to the reviewer's comments is found in the accompanying table

Reviewer 2 Report

Some or most of the authors of this paper have published several research papers on the same topic. (Ref 6,7,9,10). In my opinion, the current manuscript seem to be an incremental research and doesn't provide much insight or understanding in the advancement of science/ therapeutics. Most of the results in the paper have been explained by providing some rationale without citing proper source for such reasoning. I do not consider this manuscript to provide any additional value except for acting as a database of experiments conducted for buccal films, and thus do not recommend the publication in Materials. The authors should clearly highlight the significant progress made in this research, and how their research differs from the previous research, and further explain the plots with firm scientific insights with proper conclusions and citations.

Author Response

We wish thank to the reviewer for the comments. In the introduction of the original version the authors presented the previous works of the group. Briefly, previously we only obtain controlled release of MN when nanoparticles were developed; while the different films previously developed, presented antifungal activity only for 8h. The aim of this work was to obtain a system which allow to reduce the number of applications per day to improve the patient’s compliance and also add the effect anesthetic of lidocaine to treat the burning pain and pruritus caused by oropharyngeal candidiasis. In the R1 version the authors highlighted the significant progress made in this research. 

Reviewer 3 Report

The form of drug delivery to its destination is a very important and significant problem. The success of the treatment of the disease largely depends on this. The present article deals namely with the search for a convenient form in which the drug (here miconazole nitrate) can be delivered. The authors suggest the use of films containing an anesthetic in conjunction with an active drug. A wide range of different studies has been carried out. They are well made and I have no complaints about them. Also, the article itself is well written and interesting.

But! Oropharyngeal candidiasis is a very, very common disease that probably everyone has. There are many ways to treat this disease. And most importantly, it is really very well treated. In this regard, can the authors explain more specifically (for example, in the conclusions) why their method is better than, for example, simply wiping the affected area with cotton soaked in a solution of the drug.

Another question is the choice of the active substance. Why exactly miconazole nitrate? Why not the clotrimazole or something else?

It would also be interesting to see how such a film looks like, what is its size, for example. Is it possible to provide a photo?

The text of the article also contains a small number of typos and not very successful expressions (for example, lines 28, 53, 67, etc.)

However, I believe that the article can be published if the authors make some revisions and additions to it.

Author Response

(The authors gave the same response as above.)

Round 2

Reviewer 1 Report

The new version of this manuscript is greatly improved than the first one. Introduction and abstract are now well written. I Appreciate that Authors have considered seriously the comments and modified the work taking them into account. However, overall, the manuscript remains difficult to read because of a poor English used. Many typos also need to be fixed (e.g. surname in reference 5).  As a consequence, I strongly suggest reviewing the whole work from a stylistic, grammatical and syntactical point of view to improve its form and make it suitable for publication.

Author Response

We wish thank to the reviewer for her/his comments.

Reference 5 was modified, and the whole work was revised

Reviewer 2 Report

The authors have ignored my remark on "further explain the plots with firm scientific insights with proper conclusions and citations. "

For example: The authors have explained most of the observations using words such as "probably" (Line 251 - 254, 301-303, 383-385 (strongest interaction -- how do the authors know that?)), "could be" (Line 262 - 265, 356 - 360), Line 265 - 268 (CH -SLS have less free charges), Line 396-398 (High Charge density of both polymers and large number of free sites ?)

Please add proper citations that explain the basis/reasoning for such claims. If there are no proper citations available, the authors should make an effort to provide the chemical/physical insights that led them to come to such conclusions/reasoning. 

The authors addressed my novelty concern by stating "While the different films previously developed, presented antifungal activity only for 8h. The aim of this work was to obtain a system which allow to reduce the number of applications per day to improve the patient’s compliance and also add the effect anesthetic of lidocaine to treat the burning pain and pruritus caused by oropharyngeal candidiasis." However, only the last figure (Fig. 8) shows 24 h duration. The other figures involving duration (Fig. 1, 7 ) are shown till 400 mins. What was the reason for not showing Fig. 7 for the total duration of 24 hr?  If the curves saturate before reaching 24 hrs, what is the purpose of keeping the buccal film on for 24 hrs?

In Fig. 4a, the peak is at 300 deg C. In Fig.4b, the different film formulations also peak at ~ 300 deg C. However, the authors' comment (Line: 316-318): " In both cases, the disappearance of the drug degradation peak may have been due to the thermal protection provided by the polymers to the drugs, or that both LDCH and MN reduced their crystallinity or changed to an amorphous state". I would request the authors to elaborate on this comment.

 Also, the authors should spend significant time proofreading the manuscript and correcting the grammatical errors and typos. 

Grammar-related issues:

Line 34, 42-43,64-65, 107-108,145-146,325-328,  

Author Response

We thank the reviewer for her/his comments. Some experiments were repeated as suggested by the reviewer. Additionally, we have tried to improve the language of the manuscript. In the attached file there is a table where your comments are answered point by point

Round 3

Reviewer 2 Report

I recommend the manuscript for publication. Please proofread the manuscript for English and grammar-related issues.